# A random-walk benchmark for single-electron circuits

David Reifert [1], Martins Kokainis [2,3], Andris Ambainis[2], Vyacheslavs Kashcheyevs [3] & Niels Ubbelohde [1✉]

Mesoscopic integrated circuits aim for precise control over elementary quantum systems. However, as fidelities improve, the increasingly rare errors and component crosstalk pose a challenge for validating error models and quantifying accuracy of circuit performance. Here we propose and implement a circuit-level benchmark that models fidelity as a random walk of an error syndrome, detected by an accumulating probe. Additionally, contributions of correlated noise, induced environmentally or by memory, are revealed as limits of achievable fidelity by statistical consistency analysis of the full distribution of error counts. Applying this methodology to a high-fidelity implementation of on-demand transfer of electrons in quantum dots we are able to utilize the high precision of charge counting to robustly estimate the error rate of the full circuit and its variability due to noise in the environment. As the clock frequency of the circuit is increased, the random walk reveals a memory effect. This benchmark contributes towards a rigorous metrology of quantum circuits.

[1] Physikalisch-Technische Bundesanstalt, Braunschweig 38116, Germany. [2] Faculty of Computing, University of Latvia, 19 Raina Boulevard, Riga LV-1586, Latvia. [3] Department of Physics, University of Latvia, 3 Jelgavas Street, Riga LV-1004, Latvia. ✉email: niels.ubbelohde@ptb.de

Precise manipulation of individual quantum particles in complex single-electron circuits for sensors, quantum metrology, and quantum information transfer[1,2] requires tools to certify fidelity and establish a scalable error model. A similar challenge arises in the gate-based approach to universal quantum computation[3–8] where benchmarking gate sequences[9–13] are employed to validate independent-error models[14] which are crucial for scaling towards fault-tolerance[15,16]. Here, we introduce the idea of benchmarking by error accumulation to integrated single-electron circuits. We experimentally realize the clock-controlled transfer of electrons through a chain of quantum dots, and describe the statistics of accumulated charge by a random-walk model. High-fidelity components and unprecedented accuracy of charge counting enable the detection of excess noise beyond the sampling error, the identification of the timescale for consecutive step interaction, and an accurate estimate for the failure probabilities of the elementary charge transfer. Abstracting errors from component to circuit level opens a path to leverage charge counting for microscopic certification of electrical quantities challenging the precision of metrological measurements[17], and to introduce fidelity control in building blocks of quantum circuits[18–21].

In quantum metrology, stability and reproducibility of the environment for elementary quantum entities (photons, qubits, electrons) and their uncontrolled interactions set the practical limits on the precision of quantum circuits[22], which approach the fundamental quantum limits, i.e., counting shot noise for independent identical particles, or the Heisenberg limit for entanglement-enhanced measurements[23]. In particular, accurate benchmarking of fidelity in the presence of long-term drifts and memory is difficult but essential for the validation of the precision of quantum standards. Identifying and quantifying the residual error, i.e., any deviation from the perfect performance of a circuit, define the challenge to be answered by the random-walk benchmarking for high-precision single-electron current sources. Validating consistency of the error model by statistical testing ensures the robustness of the fidelity estimates, which is an actively studied problem in the related context of assessing quantum computation platforms[14,24–26].

The random-walk benchmarking addresses the question of uniformity in time of repeated identical operations by error accumulation. The error signal (syndrome) considered here is the discrete charge stored in the circuit after executing a sequence of $t$ operations. The measured deviation $x$ in the number of trapped electrons is modeled by the probability $p_x^t$ for a random walker to reach integer coordinate $x$ from initial position of $x = 0$ in $t$ steps (Fig. 1). In the desired high-fidelity limit of near-deterministic on-demand transfer of a fixed number of electrons any residual randomly occurring errors that alter $x$ will be very rare and the walker will remain stationary most of the time, with occasional steps of length one. Here we study to what extent two single-step, $x \rightarrow x \pm 1$, probabilities $P_\pm$ describe the statistics of $x$ collected by repeated operation of the circuit, and how deviations from independent-error accumulation can be detected and quantified, revealing otherwise hidden physics. The baseline random-walk model with $t$- and $x$-independent $P_\pm$ predicts the following distribution:

$$p_{x \geq 0}^t = \left(1 - P_+ - P_-\right)^{t-x} \left(P_+\right)^x \binom{t}{x} \times$$
$$_2F_1\left(\frac{x-t}{2}, \frac{x-t+1}{2}; x+1; \frac{4P_+P_-}{\left(1-P_+-P_-\right)^2}\right) \quad (1)$$

with $p_{x<0}^t$ obtained from Eq. (1) by $x \rightarrow -x$ and $P_\pm \rightarrow P_\mp$ (see derivation in Supplementary Note 1). Here the first term of the product describes the decay of fidelity that is exponential in $t$, while the binomial coefficient and the Gaussian hypergeometric

function $_2F_1$ (here a polynomial of order at most $t$) take into account the self-intersecting paths as single-step errors accumulate and partially cancel at large $t$ (Fig. 1a).

## Results

**A circuit for transferring electrons with high fidelity.** Experimentally, the high-fidelity circuit for electron transfer is realized by a chain of quantum dots in which the first and the last dot are operated as single-electron pumps[27] and the central dot provides the error signal as shown in Fig. 1. A clock of frequency ($f = 30$–$300$ MHz) drives the pumps to transfer one electron per cycle through the chain (from top to bottom in Fig. 1).

Within one clock cycle, the entrance barrier to the dynamic quantum dot is lowered and raised by the pump stimulus, isolating one electron from the source reservoir and then ejecting it over the high exit barrier; barrier height asymmetry between entrance and exit defines the transfer direction[28]. The operating points of the pumps are chosen to minimize and approximately balance the error probabilities of transferring either zero or two electrons instead of one (with a slight bias towards zero-electron transfers, as this error rate only increases exponentially and not double-exponentially with deviations from the optimal operating point[29]). The working points of the pumps are not retuned when operating the full circuit. Reproducible formation of quantum dots[30] allows demonstrating the high-fidelity operation of the circuit event at zero magnetic field, at which readout precision is enhanced by cryogenic reflectometry.

The excess charge $x$ from accumulating errors is inferred from a differential measurement by a charge detector capacitively coupled to the central dot, reading out the detector state before and after each sequence transferring $t$ electrons. As tunneling events are only enabled by the clocked stimulus applied to the pumps, a long detector integration-time up to 1 ms can be chosen for unambiguous identification of $x$ with a signal to noise ratio of 17 (Fig. 1b). A full histogram of detector states before and after the transfer sequence allows to reconstruct the shape of the Coulomb blockade peak resonance utilized by the charge detector and provides rigorous classification thresholds for the identification of $x$. The sequence of electron transfer and charge detection is repeated with the repetition rate limited by the detector integration-time (up to 4 kHz), until a set number of counts ($N = 1 \times 10^5$ to $2 \times 10^6$) is accumulated. Any deviations not aligned with the measurement timing, such as instabilities in the charge detector, are readily recognized and discarded, while unintended charge transitions during the operation of the pump are counted and correctly identified as errors.

Although the individual accuracy of the active components can exceed metrological precision[31], their simultaneous operation in a mesoscopic circuit[32] precludes the prediction of transfer fidelity from component-wise characterization due to interactions and crosstalk between the elements in the chain, exemplifying the need for circuit-level benchmarking. Experimental evidence for strong discord between component-wise and circuit-level characterization is given in Supplementary Note 3.

**Implementation of a random-walk benchmark.** Here we report the measurement results on two devices: device A introduces the methodology to resolve effects beyond statistical noise of independent-error accumulation in a high-fidelity circuit, while device B demonstrates the effects of memory with increased repetition frequency. Both devices share very similar device geometries and parameters.

Figure 2a shows the counting statistics measured for device A at $f = 30$ MHz for $t$ up to $10^4$ compared to predictions of the baseline model. General trends expected from the random

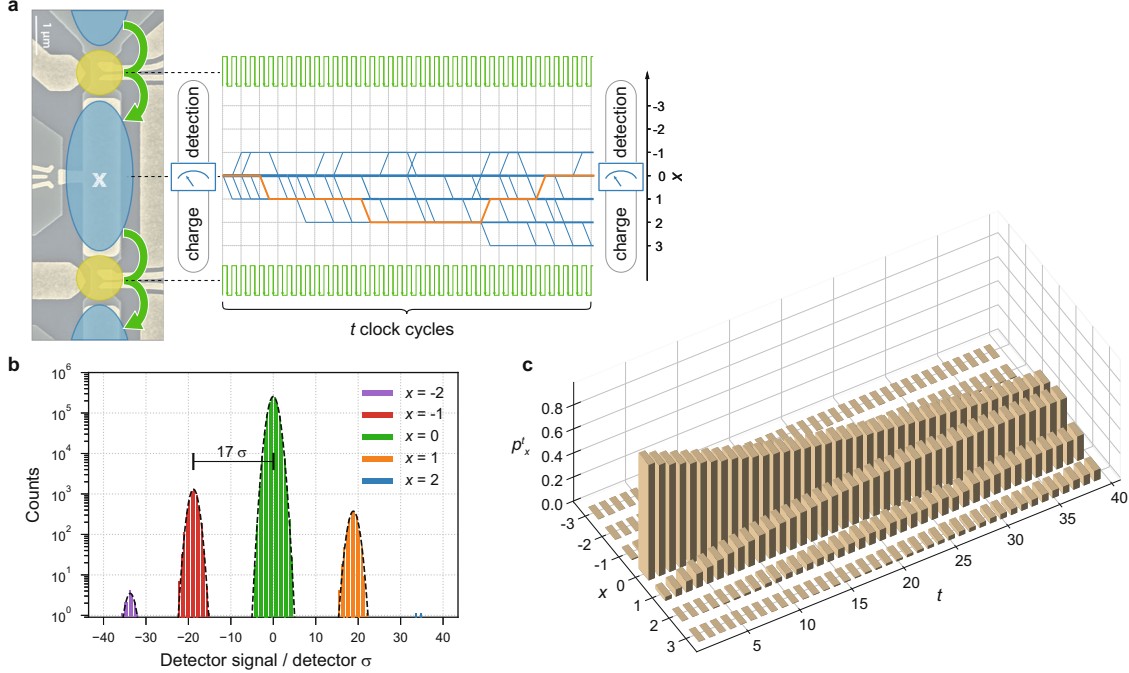

**Fig. 1 Schematic of the random-walk benchmark for single-electron transfer utilizing charge counting. a** Sample micrograph and measurement scheme. After the initial charge measurement $t$ clock cycles are applied. The paths taken by 30 simulated walkers (using error rates extracted from the counting statistics) are represented by blue lines, transitioning every clock cycle in $x$ by a step of $-1, 0, +1$. The frequency with which each branch is visited is indicated by the linewidth. A final charge measurement yields the end-point of the random walk as the difference between initial and final charge. The orange line exemplifies a single random walk with self-intersections. **b** Signal to noise ratio: a (typical) histogram of the differential charge detection signal with the identified difference in electron number indicated by color. The peak separation is shown in units of the Gaussian noise amplitude $\sigma$ (black dashed lines indicate the corresponding Gaussian fits). **c** Measured statistics of finding the walker at position $x$ after $t$ steps.

walk are evident: for short sequences, $t < 1000 \ll (P_+ P_-)^{-1/2}$, the power-law rise of the probabilities $p^t_{|x|>0}$ corresponds to the exponential decay of error-free transfer fidelity $p^t_0$, which remains close to 1. For longer sequences the distribution spreads and the weight of self-intersecting paths (e.g., orange line in Fig. 1) increases, in accordance with Eq. (1).

The key question for random-walk benchmarking is whether the uncorrelated residual randomness defined by two probabilities $P_+$ and $P_-$ predicts the entire probability distribution. This question is answered in three steps: (i) significance testing of deviations from the baseline model as a statistical null-hypothesis to delineate the inevitable sampling error from model error; (ii) extending the model to accommodate correlated excess noise[33] detected in the first step; (iii) perform parameter estimation of the noise model that yields average values of $P_\pm$ with an estimate of the variability.

For consistency testing, we have increased the number $N$ of samples per sequence by a factor of ~10, and limited $t$ to 100. Fisherian significance tests[34] are used to define consistency regions of $p$-value $> 0.05$ in the parameter space $(P_+, P_-)$ where the baseline model cannot be rejected at this significance level (see "Methods" section). Figure 2b shows quasielliptic consistency regions computed for each sequence length $t$ separately, randomly clustering in a tight area with the sizes shrinking roughly as $\sim 1/\sqrt{t}$, as expected. Their overlap is only partial: best-fit global $(P_+, P_-)$ estimated from maximal likelihood (marked on the axes of Fig. 2b) lies outside of 7 regions out of 42. A more rigorous test on whether this inconsistency can be explained by sampling error alone is provided by Fisher's meta-analysis method (Fig. 2c): under the null-hypothesis, the cumulative distribution of $p$-values obtained separately for each sequence length $t$ should be uniform (a straight line)[35,36] (Supplementary Note 6), which is not the case for the best-fit baseline model (triangles in Fig. 2c). Quantitatively, the baseline model yields global Fisher's combined $p < 3 \times 10^{-6}$, and hence is statistically rejected. We attribute this incompatibility to excess noise due to imperfections in the physical realization of the baseline model. Nevertheless, the partial overlap and the tight clustering observed in Fig. 2b suggests that the excess noise is rather small. We model the excess noise as stochastic variability of $P_\pm$, and check whether it can be plausibly explained by the presence of two-level fluctuators[37].

To quantify the excess noise, the model is now extended (part (ii) of the outline above) by drawing the step probabilities $P_\pm$ randomly from a Dirichlet distribution[38,39] (Supplementary Note 8) over the standard 2-simplex; the corresponding parameters $\boldsymbol{\alpha} = \left\{\alpha\langle P_-\rangle, \alpha(1-\langle P_+\rangle-\langle P_-\rangle), \alpha\langle P_+\rangle\right\}$ are specified by two means, $\langle P_\pm\rangle$, and one additional concentration parameter $\alpha$ which controls the variance, $\Delta P^2_\pm = \langle P_\pm\rangle(1-\langle P_\pm\rangle)/(\alpha+1)$. The Dirichlet distribution is strongly peaked near the mean point for $\alpha \gg \langle P_\pm\rangle^{-1}$, and always guarantees $0 \leq P_\pm \leq 1$. This extra randomness can be introduced at different timescales[24]. Uncorrelated noise (new $P_\pm$ after each step of a walk) is equivalent to the baseline model with $P_\pm \rightarrow \langle P_\pm\rangle$, and is already ruled out by the significance tests above. We compare a "fast fluctuator" model in which a new pair of $P_\pm$ is drawn independently after completion of each individual random walk versus a "slow drift" model in which the values of $P_\pm$ are randomly reset only after all $N$ realizations for a fixed number of steps have been collected (precise excess noise model definitions are given in Supplementary Notes 9 and 11, and the data acquisition timeline is illustrated in Supplementary Fig. S1). Although short of proper time-resolved noise metrology[26], contrasting these two correlated-noise models gives an indication of the relevant timescales (nanoseconds versus half-hour in the

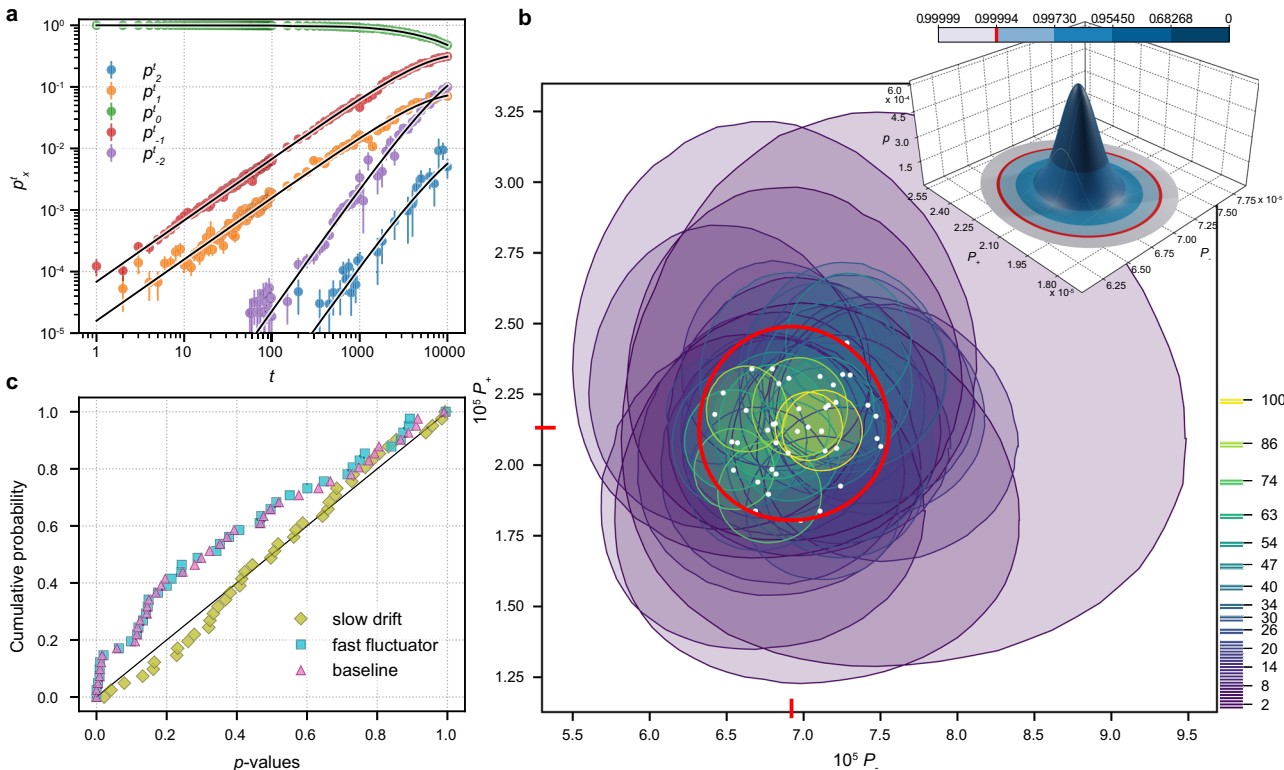

**Fig. 2 Consistency of random-walk models with respect to the experimental statistics of accumulated errors. a** Measured $p_x^t$ for device A; error bars are given by the standard deviation of the binomial distribution, solid lines show a least-squares fit of Eq. (1). **b** Likelihood-maximizing $P_\pm$ (white dots) and $p > 0.05$ consistency regions estimated separately for each sequence length (coded by color). The inset shows the probability density function of the Dirichlet distribution with parameter $\alpha = (2.43 \times 10^3, 3.50 \times 10^7, 7.47 \times 10^2)$. The corresponding global best-fit values for $P_\pm$ are marked by red lines on the axes of the consistency-region plot. The color scale indicates the level of confidence at different coverage factors $k$ for a symmetric normal distribution; the red circle in both plots and the marker in the color scale indicates the region corresponding to $k = 4$. **c** Empirical cumulative distribution of $p$-values for different models in comparison to the uniform distribution (black line).

experiments). The sensitivity of Fisher's significance testing makes it possible to distinguish between the two models, which cannot be resolved by the second moment of $\langle p_x^t \rangle$ as utilized, e.g., for noise-averaged fidelities in randomized benchmarking of quantum gates[33]. The results of Fisher's combined test (Fig. 2c) favor the "slow drift" ($p = 0.71$) over the "fast fluctuator" ($p < 3 \times 10^{-6}$) model. The corresponding best-fitting Dirichlet distribution (parameters indicated by red lines on the axes of Fig. 2b and plotted in the inset) gives $1\sigma$ uncertainty estimates $P_- = (6.92 \pm 0.14) \times 10^{-5}$ and $P_+ = (2.13 \pm 0.08) \times 10^{-5}$. Parametric instability at only a few-percent level validates a suitably extended random-walk model as a robust representation of error accumulation in this high-fidelity single-electron circuit.

In order to gain insight into a possible physics mechanism for excess noise and illustrate the robustness of statistical methods, we have simulated the experimental timeline using a random-walk model with $P_\pm$ parameters subjected to $1/f$ noise from an ensemble of independent two-level fluctuators (Supplementary Note 13). The results follow the general pattern outlined above: (i) for a fixed size of the statistical sample, there is a threshold in the excess noise amplitude above which the data contradict both the baseline and the fast-fluctuator models but remain consistent with the slow-drift model. This threshold corresponds to excess noise sufficiently affecting probabilities of multiple errors per burst to reveal inconsistency with Eq. (1) in the tails ($|x| > 1$) of the error syndrome distribution $p_x^t$. (ii) The estimated best-fit $\Delta P_\pm$ parameters correlate well with the standard deviation of the $P_\pm$ in the underlying simulation. (iii) Even a single fluctuator with a

fixed switching rate (bimodal distribution of $P_\pm$ and a Poisson distribution of switching times[40]) can generate detectable excess noise still consistent with our Dirichlet-based statistical models. As for the physics of the real device in a noisy environment, the simulations favor an explanation of the detected excess noise by the presence of multiple charge fluctuators over a single two-level system due to the absence of a bimodal signature in Fig. 2b. In conclusion, accurate statistics of error counts can give enough sensitivity to reliably estimate the baseline error rates $P_\pm$ and even capture a fingerprint of long-time correlations in the environment.

**Benchmarking for memory effects.** The methodology to quantify independent-error accumulation described above makes it possible to probe the effect of increased clock frequency on the circuit and thereby investigate response times of the electron shuttle and interactions between subsequent steps. In device B, the error rates are $P_- = (6.31 \pm 0.23) \times 10^{-3}$ and $P_+ = (2.71 \pm 0.043) \times 10^{-2}$ at the same frequency of 30 MHz as device A investigated above. A ten-fold increase of the clock frequency to 300 MHz is introduced by uniform time compression of signals controlling the transfer operations; the resulting counting statistics is presented in Fig. 3a (circles). The random-walk model with constant $P_\pm$, described by Eq. (1), no longer applies even qualitatively, which raises the question whether the fidelity of the circuit has decreased to a point where errors can no longer be considered rare as outlined in the beginning. This question is answered in the negative with the help of the following theorem

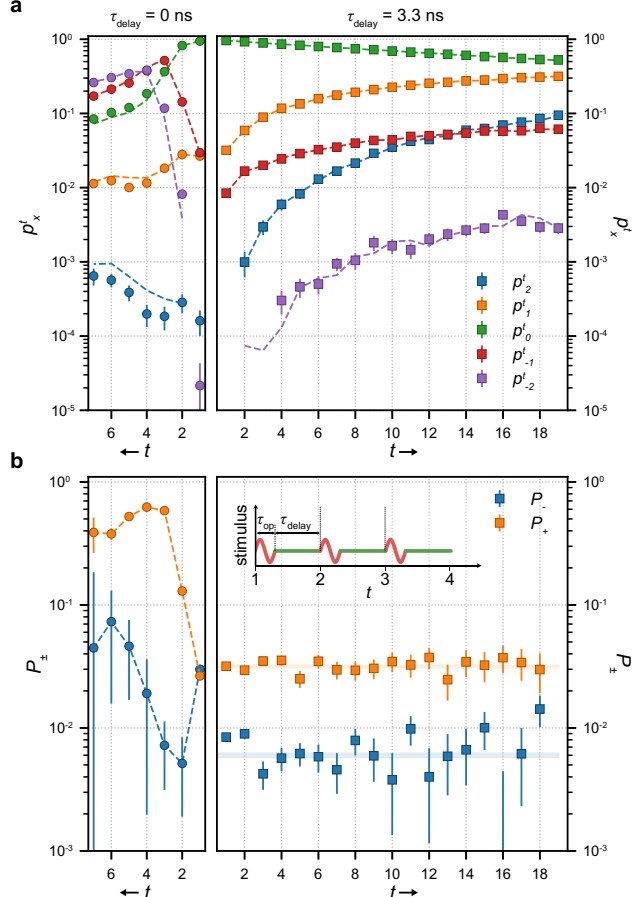

**Fig. 3 Memory effect probed at the increased clock frequency. a** Measured $p_x^t$ for device B at a clock frequency of 300 MHz and $\tau_{\text{Delay}} = 0$ s (left, $t$-axis inverted) and $\tau_{\text{Delay}} = 3.\overline{3}$ ns (right). Dashed lines represent $p_x^t$ predicted by deconvolved single-step error rates and $p_x^{t-1}$. **b** Single-step error rates $P_{\pm}^t$ for $\tau_{\text{Delay}} = 0$ s (left, $t$-axis inverted, dashed lines show guide to the eye) and $\tau_{\text{Delay}} = 3.\overline{3}$ ns (right, translucent area corresponds to the 1 $\sigma$ uncertainty estimates). Inset depicts the timing diagram of the sequence —a stimulus of duration $\tau_{\text{op}}$ drives the transfer operation followed by delay time $\tau_{\text{Delay}}$ before the next step.

defining a spread condition, which sets a precise bound on the applicability of the random-walk approach with possibly non-stationary error rates: If distributions $(p_x^t)$ and $(p_x^{t+1})$ satisfy

$$\sum_{y=-\infty}^{x-1} p_y^t \leq \sum_{y=-\infty}^{x} p_y^{t+1} \leq \sum_{y=-\infty}^{x+1} p_y^t \quad \text{for all } x, \qquad (2)$$

then there exists a set of transition probabilities $P_{\pm 1}^{(x,t)}$ such that $(p_x^{t+1})$ is generated from $(p_x^t)$ by a Markov chain $p_x^{t+1} = p_x^t + \sum_{s=\pm 1} \left[ P_s^{(x-s,t)} p_{x-s}^t - P_s^{(x,t)} p_x^t \right]$. Conversely, any discrete-space, discrete-time random walk with steps of lengths at most 1 (our definition of a high-fidelity circuit) satisfies the spread condition (2), see Supplementary Note 16 for proof of both claims.

We find that the distributions measured on device B do satisfy the spread condition (2) as long as all $x$ are fully resolved in counting ($t \lesssim 6$). We estimate the non-stationary but $x$-homogeneous single-step error probabilities of the corresponding Markov chains, $P_{\pm 1}^{(x,t)} = P_{\pm}^t$, by a numerical deconvolution of the Markov process equation (Supplementary Note 2). The resulting error rates $P_{\pm}^t$ in Fig. 3b provide reasonable prediction (dashed

lines) of the measured $p_x^t$ in Fig. 3a (circles). The $t$-dependence of $P_{\pm}^t$ is strong and reproduced well above the noise. This implies memory: probabilities for the next step depend on how many steps have taken place before. $P_{\pm}^t$ do not saturate within $t \lesssim 6$ indicating a long memory time of more than $6 \tau_{\text{op}} = 20$ ns.

To probe this memory effect, we introduce a delay time $\tau_{\text{Delay}}$ between otherwise unaltered signals driving the transfer operations thus extending the physical time $f^{-1}$ corresponding to a single step of the random walk from $\tau_{\text{op}}$ to $\tau_{\text{op}} + \tau_{\text{Delay}}$ as sketched in Fig. 3b. With increasing delay, a gradual reduction of the $t$-dependence in $P_{\pm}^t$ is observed until, for $\tau_{\text{Delay}} > 3$ ns (see right part of Fig. 3a, b), the stationary behavior consistent with the baseline model is recovered. Surprisingly, $\tau_{\text{Delay}}$ sufficient to recover stationary behavior is on the order of a single-step duration $\tau_{\text{op}}$, significantly shorter than the number of steps with pronounced memory effect at $\tau_{\text{Delay}} = 0$ ns (Fig. 3b). Both times are significantly longer than the expected timescales in GaAs systems for relaxation via electron–electron or phonon interaction[41–43], and raise the need for a dedicated investigation. In Fig. 3b, $P_{\pm}^t$, estimated at each $t$ by deconvolution (squares), are compared with the confidence intervals of the "slow-drift" model with stationary $P_{\pm}$ (color bands). The comparison shows good agreement and is consistent with our framework for random-walk benchmarking of high-fidelity single-electron circuits. For the showcased device, circuit-level interactions and memory effects significantly lower the attainable clock speed compared to record frequencies for individual pumps reported in the literature[44]. However, benchmarking by error accumulation introduces a tool to investigate these limitations and identify possible mitigation-techniques since $\tau_{\text{op}}$ and $\tau_{\text{Delay}}$ can be freely adjusted with error rates still accurately estimated on the circuit level, as long as these remain within the high-fidelity bound monitored by the spread condition.

In conclusion, the view of single-electron components as elements of a digital circuit has enabled an abstract and universal description of fidelity in terms of the random walk of an error syndrome. Accumulation of errors over long sequences allows probing fast and accurate operations beyond the bandwidth of a slow single-charge detector. The accompanying statistical methodology quantifies the stability of the error process and uncovers short memory times, both of which are elusive to direct observation. In quantum metrology, an accurate estimate of the circuit error has an immediate application: the variance of the current $I = (I_s + I_d)/2$ flowing into ($I_s$) and out of ($I_d$) the circuit is given by the variance of the differential charge $x$, which corresponds to the displacement current $I_s - I_d = efx/t$. Hence, the variance of $x$, $\Delta x^2 \approx (\langle P_+ \rangle + \langle P_- \rangle) t + (\Delta P_+^2 + \Delta P_-^2) t^2$, provides a bound for the deviation of the current $I$ from the error-free value $ef$, enabling counting-verification of a primary standard for the ampere. In the broader context, sensitive tests of single-electron circuits create new ground for developing benchmarking techniques of engineered quantum systems.

## Methods

**Devices**. Devices A and B were fabricated from GaAs/AlGaAs heterostructures with two dimensional electron gas (2DEG) nominally 90 nm below the surface. Quantum dots are formed by CrAu top gates depleting a shallow-etched mesa[30]. The charge detector is formed against the edge of a separate mesa and capacitively coupled to the central quantum dot via a floating gate[45].

**Measurement setup**. All measurements were performed in a dilution refrigerator at a base temperature of 20 mK and 0 T external field. The charge detector signal is readout by rf reflectometry[46]. Sinusoidal pulses generated by arbitrary waveform generators modulate the entrance barriers of the single-electron pumps and drive the clock-controlled electron transfer[27]. The drift-stability due to control voltages is estimated to be better than $10^{-8}$. Charge transfer and detector readout are triggered in a sequence: (i) readout of the initial detector state, (ii) application of $t$

sinusoidal pulses to both pumps simultaneously, (iii) readout of the final detector state, (iv) reset by connecting the intermediate dot to source. The difference between initial and final detector state yields the charge $x$ deposited on the central quantum dot by the burst transfer, providing raw data for subsequent statistical analysis.

**Consistency testing.** Fisher's $p$-value for each experimentally measured $x$-resolved set of $N$ counts is defined as the probability of an equally or more extreme outcome under the null-hypothesis being tested (either the baseline random walk or one of the two excess noise models with Dirichlet-distributed $P_\pm$); it is evaluated by Monte Carlo sampling as described in the Supplementary Notes 4 and 8.

## Data availability
The data that support the graphs of this work are available in the Zenodo repository https://doi.org/10.5281/zenodo.4287363.

## Code availability
The code producing the figures is available from the corresponding author upon reasonable request.

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

## Acknowledgements
We acknowledge T. Gerster, L. Freise, H. Marx, K. Pierz, and T. Weimann for support in device fabrication, J. Valeinis for discussions. D.R. additionally acknowledges funding by the Deutsche Forschungsgemeinschaft (DFG) under Germany's Excellence Strategy—EXC-2123 —90837967, as well as the support of the Braunschweig International Graduate School of Metrology B-IGSM. M.K., A.A., and V.K are supported by Latvian Council of Science (grant no. lzp-2018/1-0173). A.A. also acknowledges support by 'Quantum algorithms: from complexity theory to experiment' funded under ERDF program 1.1.1.5.

## Author contributions
D.R. and N.U. designed and performed the experiment. M.K., A.A., and V.K. developed random-walk modeling and statistical methodology. DR., N.U., M.K., and V.K. performed the data analysis. M.K. wrote the supplementary information with contributions by D.R., V.K., and A.A. All authors contributed to the discussion of results and the writing of the manuscript.

## Funding

## Competing interests

The authors declare no competing interests.
