## [Peer Review File · Nature Communications]

REVIEWER COMMENTS

Reviewer #1 (Remarks to the Author):

The paper describes the counting and statistical analysis of errors in single-electron circuits, where the transfer errors of two single-electron pumps in series are accumulated so that they give random-walk type fluctuations of electron numbers in central charge dot between the two pumps. While the device and the error counting scheme used here are similar with those in ref. [38], this work features fast charge detection by radio frequency (rf) reflectometry and statistical hypothesis testing based on measured statistics of accumulated errors. Rigorous adaptation of so called Fisher's method reveals that the error rate is not constant as simply expected, but it can vary with some slow drift or memory effect. The paper is nicely written, addressing such detailed nature of errors in single-electron devices with sound statistical treatment. The analysis shown here is impressive by use of rigorous adaptation of statistical testing. There is no doubt that the present study is at a high level in terms of experiments and mathematics. I believe that this paper should be worth publications for good quality journals. However, I have seen that the impact of the work is weak in the following points and therefore feel that it would not reach the level required for Nature Communications.

- i) The usefulness of the presented method for quantum computer technology or future quantum system is not very clear. I wonder how much the presented scheme is demanded from such related community and how significantly it can be compared with the randomized benchmarking widely used for quantum gates. A small drift of the error rate discovered in this work would be not a major limiting factor for the present state-of-the-art quantum computers.
- ii) To my understanding, the used method for the statistical test itself is not new.
- iii) Experimental finding of some small drift or memory effect in the pump error is not very surprising. One can easily imagine that such an instability may happen in real devices.
- iv) From the metrology viewpoint, pump error demonstrated here (order of 10^{-5}) is not low compared to previously reported ones.

Below are several points I would like to make comments on the manuscripts.

1) The authors use a Dirichlet distribution (continuous model) probably because it is easy to handle from the mathematics viewpoint, but I wonder how this method is suitable in terms of the two-level fluctuator (discrete and bimodal) model mentioned in the main text. I see that there is still some deviation between the straight line and the slow drift seen in Fig. 2 (c). Does this mean something related to the limit of the model? Furthermore, while the presented analysis model introduces a kind of artificial and instant fluctuation in time domain, one can also expect well known $1/f$ noise in devices. I would like the authors to comment on how differently the test statistic can behave in the combined p-value analysis for such a more realistic case.

2) The instability can also exist in the charge detector. I would like the authors to discuss the possibility that the observed drift/memory effect is related to it and explain why it can be excluded.

3) The device consists of two single-electron pumps and the variance of the generated current is dominated by the one with a larger error rate, which would be a drawback for realizing the practical quantum current standard with the present device configuration because the yield of the high-accuracy pump would be not so high. Actually the reported errors of 10^{-5} and 10^{-2} are far from metrological requirement. I would like the authors to address to strategies for the route to the realization of a primary current standard.

4) I notice that P^+ and P^- differs in both A and B devices. I wonder why these conditions were chosen. The operating point can be tuned by the gate voltage in such a way that the errors are symmetric or the variation of the pump current is minimized. For the memory effect in device B with $T_{\text{delay}}=0$, both

P+ and P- decrease with time, which seems to suggest that successive steps reduce the error, perhaps by a kind of noise reduction. On the other hand, it is shown that the intermittent steps with nonzero Tdelay give small P+ and P- as well. I wonder how these two behaviors can be consistent.

Reviewer #2 (Remarks to the Author):

In this work, the authors investigate a single-electron circuit, made of quantum dots through which single electron tunneling events take place. In the spirit of benchmarking methods widely used to develop scalable error models in quantum information processing, they develop a benchmarking method for characterizing single-electron charge transport with tools and quantities suitable to developing metrological standards for the current for instance.

The topic is highly interesting, challenging and relevant for several fields of research, and propose a novel research direction combining single-electron based circuits and scalable error models developed for information processing. To my opinion, this interdisciplinary work is highly original and I would definitely recommend its publication in Nat. Comm.

However, before, I would recommend the authors to consider the following questions / comments / suggestions. I believe that the manuscript would gain in clarity and would be more accessible to a wide audience if some points are reshuffled. I will list below my comments in order of appearance, but before, my main suggestions are:

A) to better convey at the beginning of the work the idea that the experiment and the analysis can be thought within the context of quantum metrology as nicely explained in the conclusion. This paragraph, to my opinion, should appear much earlier in the work.

B) to better explain the experiment. Although being familiar with single-electron circuits, figure 1 and the text are most probably not sufficient to make clear what is the experiment. I will provide more details below. This will also allow the authors to introduce, motivate and better justify the random walk model they use for their benchmarking.

Ps: my list is long, but it is because I really like the paper and the work!

List:

1) After the intro, the authors directly goes to their model, the baseline random-walk model. I would definitely explain in few words the experiment and then say that you suggest the distribution (1) derived from a multinomial distribution assuming a random walk. I went through the appendix I (see comments below) and although you provide their the details for deriving (1), I think it's helpful to at least mention key arguments such that the reader understand why you consider this model (for instance, independent jumps -> random walks, multinomial distribution etc...).

2) Related to the description of the experiment, it is not clear to me whether there is a well-defined direction for a current (source and drain) that may explain the arrows going from top to bottom or not. I understand that what is measured is the occupancy of the central dot, and electrons can arrive or leave this central dot from above or below. What do the driving potentials sequence look like? In the current form of the paper, it's difficult for me to see how the single-electron pumps and the central dot work together to achieve this random-walk like charge occupancy.

3) In Fig. 1b (a bit small to my opinion), what are the tilted bars, sometimes to the right, sometimes to the left?

4) Considering fermionic systems without a spin degree of freedom, ruled by the Pauli principle, how can the central dot be the place for reaching coordinate x in t time steps? As far as I understand, and it seems to agree with your charge detection process, you only measure the values $-1, 0, +1$, for one electron leaving, nothing happening and one electron jumping in (from top and/or bottom?). I apologize for this probably naïve question, but I am missing the argument.

5) When discussing the random-walk model page 1, you write that “any residual randomly occurring errors will be very rare...”. What are / what could be these residual randomly errors? Ps: I think that these types of questions would be easily answered if the experiment and link to the model are better explained from the beginning, see comment above.

6) Page 4: device A is introduced, but not clearly defined with respect to device B mentioned later. A bit confusing for the reader.

7) Figure 2b: If I understood correctly, the red lines on the x and y axis mark the best-fit for $P+$ and $P-$. Is this best fit done with respect to the 10 setups over $t=100$ cycles? What is the red circle? Where does it come from? Is it just to visualize the region where all your measurements lie? I also do not see clearly where these white dots come from? Measured? Estimated from the final charge detection you make on the central dot? I realized now that this may be related to note II in your supp. Material?

8) Figure 2c: I understand the different models, but do not understand where the data come from: do you obtain the different curves by using your charge detection data in the different models, and obtain that baseline is not linear as you would expect for this Fisher’s analysis method?

9) If I am correct, at the end of page 4), you have answered answered point i) mentioned above (baseline model under null-hypothesis is not enough), and you have now to determine how to take into account correlated excess noise, i.e. point ii). If so, could be helpful to write it clearly.

10) I do not understand your argument for saying that excess noise is probably small from Fig. 2b. Could you please explain better?

11) When introducing the Dirichlet distribution, maybe say in the main text that this distribution is “natural” (another word may be much more accurate!) when considering a multinomial distribution. Wikipedia’s article mentions that the two distributions are conjugate prior. Is this related to what you write in the note III?

12) When you introduce the slow drift and fast fluctuator models, it could be nice to provide in the appendix at least the mathematical expressions of these models, just to see formally how the different resets are done.

13) Page 6: you write that the slow-drift model is favored according to the Fisher’s combined test, and the you write that “validates the simple random walk as a robust representation of error-accumulation...”. Is this not in contradiction? If I understand correctly, it is a modified version of the random walk model, that takes into account correlations via the reset of P_{\pm} after few steps, that is correct, no?

14) Second part page 6: device B is introduced. I do not understand the difference wrt device A, see above.

15) Is the difference between A and B the increase in clock frequency? Increase of clock frequency means that the tunnel events happen more rapidly? Maybe this could also become clearer if the experiment is better described at the beginning.

16) Is there some known literature about non-stationary random walk models that could discuss the spread condition?

17) I would suggest the authors to consider putting the spread condition (called thm) in the supp. Material. It looks a bit weird to have a theorem in the middle of an experimental paper for benchmarking errors in single-electron circuits. I would rather emphasize the physics behind, the memory effect etc.

18) Is there a way to model the memory effect observed here through non-Markovian rate equations or other tools for instance?

Supp.material:

19) Note I: As the authors provide the full derivation of the multinomial distribution specific to their experiment, I would suggest to provide all details. To check their derivations, I had to go back to the definition of the multinomial distribution, and find what are the variables. I would suggest to start from the PMF

$$f(k-1, k_0, k_1; t, P_-, P_0, P_+) = \sum_K t! / (k! K_0! k_1!) P_-^{(k-1)} P_0^{(k_0)} P_+^{(k_1)}$$

that everyone understands. And then say that you define $s = k-1$.

How do you derive the upper bound for the sum over s , $(t-x)/2$?

You could add the definition of the Pochhammer's symbol, and provide the intermediate steps. This would also make the identification with the hypergeometric function easier (and you could also provide the steps).

You remark that other models have been previously used for this type of circuits. Could you discuss then the advantage of your model compared to those? Did you try to do the same type of analysis and significance test with those models?

Typos and minor things:

- Page 1: an fixed number of electrons -> a fixed number of electrons

- References to Eqs. do not always appear correctly, see eg. Note I before claim 1 (??). Check throughout the manuscript.

Reviewer #3 (Remarks to the Author):

Review of "A random-walk benchmark for single-electron circuits"

The manuscript uses a random-walk-based accumulation of errors statistical framework to probe the accuracy of charge shuttling beyond the bandwidth of the charge detector in a single electron circuit (more than one single electron device) operated as a pump. To do so, experiments are performed with two GaAs/AlGaAs single-gate ratchet pumps in series with a large island between them on which an RF-SET detects errors via a floating gate antenna. There are two main motivations for performing this analysis: 1) detector bandwidths are much smaller than the shuttling frequencies of metrological interest which prevents real-time error detection and 2) the errors of the individual single electron

devices cannot generally be used to calculate the error of the circuit where the devices are operated together.

The work is important directly for the metrological community and, as the authors mention, could have a long reach into quantum communication where these methods could impact the fidelity of initialization and readout of qubits minimally. I recommend significant revision of the manuscript before it is suitable for publication in Nature Communications. I discuss my reasons below.

Given the motivation discussed in the introduction (single electron circuit validation) for the study it seems a shame that the proof of this assertion (Tables S1 and S2) is at the end of a very long and dense supplemental information section. I suggest the authors incorporate this information into the main text or at least feature it more prominently in the supplemental.

The manuscript lacks experimental details necessary to understand the experiment. These include, but are not limited to, the detector bandwidth, measurement time, any thresholds used for distinguishing between charge states of the island, how each individual device was tuned, whether the devices were retuned together when combined for error measurements. Why was no magnetic field applied? The above experimental concerns seem important since whether or not the 'x' are fully resolved in counting makes the difference between stationary $P+$ and $P-$ and non-stationary values. In addition, there was no mention of a numerical value for N except parenthetically in the methods section (this was my first question when examining figures 1 and 2).

The authors should also discuss at more length the viability/accuracy/applicability of the modeling especially at higher frequencies. There is already an effective frequency limit of 150 MHz (300 MHz signal with 3.3 ns delay). Metrologists are certainly interested in higher frequencies than this. Beyond the spread condition, what are the limitations of this model?

There are several missing references in the supplemental and the grammar is sometimes incorrect leading to a lack of clarity.

The manuscript reads like a mathematical exercise without much physical insight. I do not suggest the authors reduce the formal mathematics but rather expand on issues like the speculation of two-level fluctuators and how these cause "slow" drift in $P+/-$. What can we say are the characteristics of this drift? The only connection to the physical devices offered is to say that the drift may be due to TLSs. While "fast" and "slow" drift modeling are performed, it seems to me that "fast" and "slow" are relative to the timescale for t steps. Is this correct? If so it seems that the meaning of "fast" and "slow" varies over the course of these experiments. How do we understand the modeling and source of drift in $P+$ and $P-$ in this context?

We appreciate that the reviewers commend the originality, rigor and importance of our work, and observe that their comments suggest a whole variety of new research directions, each worthy of a dedicated investigation: optimization of single-electron circuits for metrological precision, extending the statistical testing methods to quantum gate benchmarking, and discovering physically motivated non-Markovian models for the memory effects in mesoscopic circuits. We have made a substantial revision of the manuscript attempting to balance these different perspectives while preserving the coherence of the manuscript.

We also thank the reviewers for their helpful recommendations and criticisms, to which a point-by-point response is compiled below.

As the number of changes to the manuscript have been quite extensive, we highlight the revisions in the pdf by markup instead of a separate file listing page and line numbers. Furthermore, Figure 1 was rearranged and contains an additional panel illustrating detector precision. The labels of the horizontal axis in Figure 3.b. were also changed to better indicate the reversed direction.

Reviewer #1:

i) The usefulness of the presented method for quantum computer technology or future quantum system is not very clear. I wonder how much the presented scheme is demanded from such related community and how significantly it can be compared with the randomized benchmarking widely used for quantum gates. A small drift of the error rate discovered in this work would be not a major limiting factor for the present state-of-the-art quantum computers.

The need to detect and quantify correlation effects beyond independent error accumulation is a recognized and actively researched topic in the area of quantum computation. To cite a prominent example, the paper detailing the most advanced quantum processor to date [Google Sycamore, Nature 574, 505 (2019)], specifically emphasizes this aspect of their achievement in a dedicated section “Verifying the digital error model”: “A key assumption underlying the theory of quantum error correction is that quantum state errors may be considered digitized and localized ... Indeed, our experimental observations support the validity of this model for our processor.” ... “To be successfully described by a digitized error model, a system should be low in correlated errors.... Demonstrating a predictive uncorrelated error model up to a Hilbert space of size ⁵³ shows ...”

RB of quantum gates only provides average information and can misjudge system fidelity [Magesan et al., Phys. Rev. Lett. 106, 180504 (2011)]. The difficulty of accurately assessing fidelity in the presence of noise from a limited sample size in QC is acknowledged in the literature, e.g.

Epstein *et al.*, Phys. Rev. A 89, 062321 (2014).

Ball *et al.*, Phys. Rev. A 93, 022303 (2016).

Mavadia *et al.* npj Quantum Information 4, 7 (2018).

Error rates for digital operations derived both from quantum RB and from our RWB are randomly distributed and separating mere statistical noise from paradigm-violating correlated errors in this distribution is a shared challenge.

Utilizing the full distribution of fidelity under the assumption of independent error accumulation and testing this model statistically is, to the best of our knowledge, a new idea in this field.

While our implementation of this idea exploits the specifics of single-electron circuits, its extension to quantum RB would be a natural connecting path from application in electrical metrology to application in quantum computation. Our work will hopefully stimulate future theoretical and experimental work in this direction.

While a small drift of an error rate either in quantum computing or in an electron counting circuit might not appear as a limiting factor for the present state-of-the-art, it is crucial for judging scalability [npj Quantum Information 1, 15005 (2015)] and for discovering rare-event physics that can be a potential show-stopper to the ultimate asymptotic advantage (i.e., many-qubit computation or quantum standards).

ii) To my understanding, the used method for the statistical test itself is not new.

We have addressed this criticism in the response to 1.i above.

iii) Experimental finding of some small drift or memory effect in the pump error is not very surprising. One can easily imagine that such an instability may happen in real devices.

The influence of environment and memory have been identified as fundamental limits in quantum metrology and these effects are certainly to be expected in any quantum circuit yet often assumed to be negligible. However, because of the ubiquitous nature, it is even more important to achieve experimental observation and to develop methodology enabling identification and quantification.

iv) From the metrology viewpoint, pump error demonstrated here (order of 10^{-5}) is not low compared to previously reported ones.

The focus of this work is not to demonstrate the lowest pump error so far but to show how fidelity can be very precisely estimated independent from the details of pump mechanisms and not resting on predictions of component characterizations. The experimental conditions were therefore chosen to best utilize well-established models and high-fidelity counting, even though achieving low pump error thereby becomes more difficult. Still, compared to previous efforts, considerable improvement to the device performance under the given conditions was achieved, which for the first time was sufficient to investigate random-walk benchmarking by performing sequences up to a length of 10000 transfer operations (high-fidelity regime). Applying this methodology to circuits operating in high magnetic field can be expected to be accompanied by a significantly lower pump error.

1) The authors use a Dirichlet distribution (continuous model) probably because it is easy to handle from the mathematics viewpoint, but I wonder how this method is suitable in terms of the two-level fluctuator (discreet and bimodal) model mentioned in the main text. I see that there is still some deviation between the straight line and the slow drift seen in Fig. 2 (c). Does this mean something related to the limit of the model ? Furthermore, while the presented analysis model introduces a kind of artificial and instant fluctuation in time domain, one can also expect well known $1/f$ noise in devices. I would like the authors to comment on how differently the test statistic can behave in the combined p-value analysis for such a more realistic case.

The reviewer raises a number of important points here which we have researched and now address in the revised manuscript.

We have extended the discussion of two-level fluctuators, and consider separately the two scenarios suggested by the reviewer (a single TLF or an ensemble producing $1/f$ noise). We have set up simulations of these more realistic noise environments and applied our combined p-value analysis for the simulated counts. The results are summarized in the main text (see paragraph right before

turning to timing and memory effects on device B), and detailed in a new supplementary note (Note V), these confirm suitability of our methods for these physically motivated noise models, as long as the excess noise is small compared to baseline statistical fluctuations (standard deviation of the fluctuating P_{\pm} is smaller than the mean, so that the best-fitting Dirichlet distributions remain peaked).

The choice of Dirichlet distribution is motivated by it being a close analogue of the Gaussian distribution on the compact space of probabilities in the relevant regime and having only a single width parameter (a minimal necessary assumption to accommodate parametric noise). This is now clarified in the text (“The Dirichlet distribution is strongly peaked near the mean point for $\alpha \gg \sqrt{P_{\pm}}$, and always guarantees $0 \leq P_{\pm} \leq 1$.”)

Concerning the deviations between the straight line and the p-values distribution from the slow-drift test, we do not expect the empirical cumulative distribution of 42 p-values to be perfectly uniform, and judge the significance of these deviations by the combined p-value which is neither too low nor too large both for the experimental data in Fig. 2(c) and for (the newly added) simulation data in Supplementary Note V, c.f. Figure S3(c).

2) The instability can also exist in the charge detector. I would like the authors to discuss the possibility that the observed drift/memory effect is related to it and explain why if it can be excluded.

We acknowledge that this is an important aspect of our work which has not been properly communicated in the original manuscript.

Instabilities in the charge detector occurring at random times do not adhere to the timing of the clock-controlled detection cycles and are therefore visible as increased noise in the charge detector signal, which would switch to a different state during signal integration. Instabilities in the charge detector itself triggered synchronously with the electron transfer can be expected to yield a signal-response different from charge-fluctuations in the QD chain due to the drastically different coupling. As the charge detector response is reconstructed from a histogram of transfer events, bimodal deviations are easy to recognize. In contrast, any spontaneous changes in the charge occupation of the QD chain have to be counted as errors and are part of the fidelity considerations.

We have expanded the description of the experiment to acknowledge these points and added a panel to Figure 1 to further illustrate the detector precision

3) The device consists of two single-electron pumps and the variance of the generated current is dominated by the one with a larger error rate, which would be a drawback for realizing the practical quantum current standard with the present device configuration because the yield of the high-accuracy pump would be not so high. Actually the reported errors of 10^{-5} and 10^{-2} are far from metrological requirement. I would like the authors to address to strategies for the route to the realization of a primary current standard.

Connecting the pumps in series allows us to utilize the high precision of single charge detection via the readout of an error syndrome, which considerably offsets any disadvantage of this circuit design within the context of metrology. To introduce the methodology, the error rates of the demonstration devices are beneficial by achieving sufficient fidelity to operate for the first time sequences up to a length of 10000, while still keeping the measurement time required to observe the relevant error cases manageable. While not the aim of this work, the random-walk framework should of course also prove to be particularly useful to investigate experimental conditions under

which GaAs-based single electron pumps typically show significant improvement (e.g. magnetic field), and help to identify additional effects limiting the lowest achievable residual error.

The reproducibility and variability of single electron pumps is currently being studied in parallelization devices, and we would not necessarily agree that yield will likely limit the application of such circuits. However, all these circuit designs underline the need for rigorous validation.

A strategy towards the realization of a primary current standard is given by the outlook (the last paragraph of the main text), since the variance of the generated current based on the outlined fidelity estimate provides the necessary in-situ validation. Long sequence lengths should allow to generate measurable currents in the pA range utilizing the high component fidelities typically achieved in high magnetic field.

We expanded the description of the experiment to better explain the choice of experimental conditions and purpose of devices.

4) I notice that $P+$ and $P-$ differs in both A and B devices. I wonder why these conditions were chosen. The operating point can be tuned by the gate voltage in such a way that the errors are symmetric or the variation of the pump current is minimized. For the memory effect in device B with $T_{\text{delay}}=0$, both $P+$ and $P-$ decrease with time, which seems to suggest that successive steps reduce the error, perhaps by a kind of noise reduction. On the other hand, it is shown that the intermittent steps with nonzero T_{delay} give small $P+$ and $P-$ as well. I wonder how these two behaviors can be consistent.

In the decay cascade model, the error rate of transferring zero electrons decreases exponentially, while the error rate for two-electron transfers decreases double exponentially when approaching the optimal operating point. In the case of device A, which shows good agreement with this phenomenological description, a slight detuning towards 0-transfer errors is therefore preferable to rule out the influence of drifts of control voltages.

For the memory effect probed in device B no notable improvement by offsetting the operating point could be achieved. The error rates for $\tau_{\text{delay}} = 0$ s are actually increasing with t , as the horizontal axis in Figure 3 is inverted to highlight the initial conditions at $t=1$ that are common to $\tau_{\text{delay}} = 0$ s and $\tau_{\text{delay}} = 3.3$ ns.

In the revised manuscript, the description of the experiment has been expanded accordingly and arrows indicating the direction of the t -axis have been added to Fig. 3 b.

Reviewer #2:

A) to better convey at the beginning of the work the idea that the experiment and the analysis can be thought within the context of quantum metrology as nicely explained in the conclusion. This paragraph, to my opinion, should appear much earlier in the work.

Following this suggestion, we added a new paragraph after the first introductory paragraph, which puts our work in the context of quantum metrology and explains the need and importance of benchmarking. We prefer to leave the paragraph that the referee has commended at the conclusion of the paper, since it uses quantities accurately defined throughout the main text and also formulates the outlook for future work.

B) to better explain the experiment. Although being familiar with single-electron circuits, figure 1 and the text are most probably not sufficient to make clear what is the experiment. I will provide more details below. This will also allow the authors to introduce, motivate and better justify the random walk model they use for their benchmarking.

The description of the experiment has been substantially expanded, following the recommendations below

1) After the intro, the authors directly goes to their model, the baseline random-walk model. I would definitely explain in few words the experiment and then say that you suggest the distribution (1) derived from a multinomial distribution assuming a random walk. I went through the appendix I (see comments below) and although you provide their the details for deriving (1), I think it's helpful to at least mention key arguments such that the reader understand why you consider this model (for instance, independent jumps -> random walks, multinomial distribution etc...).

The reason for the chosen order of presentation is, that the model does not depend on any specific details of the experiment, but instead aims to offer an abstracted view in form of a random walk of accumulated errors. This approach towards a challenge found in a variety of applications forms an important part of the benchmark, where fidelity is not derived from pre-characterization of the components in the experiment (which are prone to be altered by cross-coupling for the full circuit). The metrological challenge lies in robustly estimating the fidelity of the circuit utilizing the precision of counting. Whether this agnostic approach can be valid for an actual implementation of a single electron circuit forms the first part of the paper.

The introduction of the experiment was therefore not moved, since the additions to the manuscript should help to clarify this point. While we agree that the multinomial distribution is a helpful concept to understand how one approaches the problem and arrive to Eq (1), we feel that this line of thought cannot be properly elaborated in the opening part of the paper. Instead, we have expanded the derivation in Supplementary I (proof of Claim 1 which starts with a multinomial distribution; discussion of the asymptotic behavior is streamlined).

2) Related to the description of the experiment, it is not clear to me whether there is a well-defined direction for a current (source and drain) that may explain the arrows going from top to bottom or not. I understand that what is measured is the occupancy of the central dot, and electrons can arrive or leave this central dot from above or below. What do the driving potentials sequence look like? In the current form of the paper, it's difficult for me to see how the single-electron pumps and the central dot work together to achieve this random-walk like charge occupancy.

The flow direction is enforced by the QDs at the ends of the chain which are tuned to function as non-adiabatic, tuneable-barrier single electron pumps. . Electrons on the central QD therefore can only arrive and leave during a pump cycle, with one electron arriving from above and one leaving to below for most of the cycles (error-less deterministic operation) During the charge detection phase no random tunneling events occur. The description of the experiment has been extended by the discussion of the pump operations cycle and now explicitly describes these charge transitions these charge transitions.

3) In Fig. 1b (a bit small to my opinion), what are the tilted bars, sometimes to the right, sometimes to the left?

The tilted bars depict the steps (transitions in x) of a few simulated random walks. The width of the lines and the bars indicates the number of times the particular path has been taken (steps with wider bars have occurred in multiple runs).

The figure caption has been revised to more clearly explain the random-walk graph, which has also been enlarged.

4) Considering fermionic systems without a spin degree of freedom, ruled by the Pauli principle, how can the central dot be the place for reaching coordinate x in t time steps? As far as I understand, and it seems to agree with your charge detection process, you only measure the values $-1, 0, +1$, for one electron leaving, nothing happening and one electron jumping in (from top and/or bottom?). I apologize for this probably naïve question, but I am missing the argument.

The direction of electron transfer is determined by the high exit barrier of the single electron pumps suppressing any back-tunneling events. As the entrance pump adds an electron (or erroneously zero or two electrons), the exit-pump removes an electron (or 0, 2) from the central dot. The detectable charge difference per step is therefore $-1, 0, +1$ (with a difference of -2 or $+2$ occurring after a single step being very unlikely for a circuit achieving sufficiently high fidelity, which is verified by the spread condition). With more steps in longer sequences these differences in transferred charge accumulate, resulting in extra charge x on the central dot after t steps. The expanded description of the experiment should clarify these points.

5) When discussing the random-walk model page 1, you write that “any residual randomly occurring errors will be very rare...”. What are / what could be these residual randomly errors? Ps: I think that these types of questions would be easily answered if the experiment and link to the model are better explained from the beginning, see comment above.

This paragraph introduces the circuit-level model and is therefore purposefully abstract. In this context 'residual error' encompasses any deviation in the number of electrons transferred through the monitoring node from the intended number, which results in an increase or decrease of the accumulated charge (introduced in the referred paragraph as the error syndrome).

We give a definition of the residual error in the added preceding paragraph on metrology and explain explicitly the residual pumping error later on in the now expanded description of the experiment. No changes were therefore made to the paragraph introducing the baseline model at the general level.

6) Page 4: device A is introduced, but not clearly defined with respect to device B mentioned later. A bit confusing for the reader.

In the description of the experiment, both devices, together with their purposes and differences, are now explicitly introduced.

7) Figure 2b: If I understood correctly, the red lines on the x and y axis mark the best-fit for $P+$ and $P-$. Is this best fit done with respect to the 10 setups over $t=100$ cycles? What is the red circle? Where does it come from? Is it just to visualize the region where all your measurements lie? I also do not see clearly where these white dots come from? Measured? Estimated from the final charge detection you make on the central dot? I realized now that this may be related to note II in your supp. Material?

The red lines indicate the best fit values for the complete dataset, i.e. all 42 different sequence lengths, while the white dots indicate maximal likelihood estimates done for each sequence length separately. The red circle illustrates the extent of the Dirichlet distribution in the 2D plot by showing the expanded uncertainty at coverage factor $k = 4$. A sentence explaining the red lines has been inadvertently removed in the submitted manuscript. Both main text and figure caption have been revised accordingly, clarifying the exact meaning of depicted quantities.

8) Figure 2c: I understand the different models, but do not understand where the data come from: do you obtain the different curves by using your charge detection data in the different models, and obtain that baseline is not linear as you would expect for this Fisher's analysis method?

The different curves represent indeed Fisher's combined test applied to the data of Fig. 2b using different models.

We revised the figure caption for clarification.

9) If I am correct, at the end of page 4), you have answered answered point i) mentioned above (baseline model under null-hypothesis is not enough), and you have now to determine how to take into account correlated excess noise, i.e. point ii). If so, could be helpful to write it clearly.

This is indeed correct. We have now marked this paragraph as the beginning of step (ii).

10) I do not understand your argument for saying that excess noise is probably small from Fig. 2b. Could you please explain better?

Fig 2b shows the consistency regions for baseline model parameters. In the absence of excess noise, there is a "true value" - a common point in parameter space located inside of most (naively - 95%) of the consistency regions. We observe that the regions computed with actual data (Fig. 2b) do overlap substantially (although not sufficiently - only 7 from 42 contours), hence extra variability beyond the baseline fluctuations is likely to be small.

We now explicitly refer to this argument in the sentence suggesting that the excess noise is small:

“Nevertheless, **the partial overlap** and the tight clustering observed in Figure 2b suggests that the excess noise is rather small”

11) When introducing the Dirichlet distribution, maybe say in the main text that this distribution is “natural” (another word may be much more accurate!) when considering a multinomial distribution. Wikipedia’s article mentions that the two distributions are conjugate prior. Is this related to what you write in the note III?

The two advantages of Dirichlet distribution as a generic model for unknown excess noise: 1) it adds only one additional parameter to characterize the spread; 2) P_{+} and P_{-} drawn from the distribution are guaranteed to belong to the simplex (never become unphysical, $P > 0$ or $P < 1$).

Dirichlet distribution with sufficiently large α is peaked in the vicinity of $\langle P_{+} \rangle$, $\langle P_{-} \rangle$, much like a Gaussian distribution but without a chance for out-of-bounds $P_{+/-}$. The “natural” relation between Dirichlet and multinomial distribution mentioned in Wikipedia is relevant for Bayesian estimation methods which are not employed here.

12) When you introduce the slow drift and fast fluctuator models, it could be nice to provide in the appendix at least the mathematical expressions of these models, just to see formally how the different resets are done.

Supplementary Note IV (in the revised numbering) defines the two models explicitly. In the beginning of Note IV A and B we describe the “fast fluctuator” and the “slow drift” algorithm explicitly. Corresponding mathematical expressions of the expected multinomial distribution of the number of counts in each category are given by Eqs. (10) and (11), respectively.

We have added subsection headers to the Supplementary Note IV as well as Table of contents for the whole set of Supplementary Notes to help the reader find the appropriate mathematical details.

13) Page 6: you write that the slow-drift model is favored according to the Fisher's combined test, and then you write that "validates the simple random walk as a robust representation of error-accumulation...". Is this not in contradiction? If I understand correctly, it is a modified version of the random walk model, that takes into account correlations via the reset of P_{\pm} after few steps, that is correct, no?

Here, 'simple' was not intended to refer to the baseline model, but rather to the general approach of estimating fidelity from a random walk of accumulating errors.

We rephrased the sentence to avoid this misunderstanding.

14) Second part page 6: device B is introduced. I do not understand the difference wrt device A, see above.

The two devices are now introduced explicitly in the beginning and their different purposes are explained.

15) Is the difference between A and B the increase in clock frequency? Increase of clock frequency means that the tunnel events happen more rapidly? Maybe this could also become clearer if the experiment is better described at the beginning.

The difference of device B is a higher error rate (and thereby potentially greater susceptibility to additional error mechanisms), yet it produces counts still compliant with the spread condition. As the clock frequency is increased, the time between the steps of the random walk/tunnel events is indeed reduced.

16) Is there some known literature about non-stationary random walk models that could discuss the spread condition?

We are not aware of such literature (which of course does not exclude that it exists), and give the full proof in the Supplementary Note VII (in the revised numbering). The application for data analysis is more likely to be genuinely new and useful and hence placed and discussed in the main text (see also our response to the next question below).

17) I would suggest the authors to consider putting the spread condition (called thm) in the supp. Material. It looks a bit weird to have a theorem in the middle of an experimental paper for benchmarking errors in single-electron circuits. I would rather emphasize the physics behind, the memory effect etc.

The spread condition serves an important purpose in the main text, as it formulates the condition under which the random-walk approach remains justified despite the increase of error rates with increasing operation frequency. A remarkable property of the spread condition is that it is both necessary and sufficient – it means that if it is satisfied, then there is a step-1 model of the circuit that produces the observed data, thus ensuring the validity of the description in terms of individual step probabilities regardless of the actual physics. The latter can be very complex (e.g., non-local in time, non-Markovian) and device-specific. When investigating the memory effect, the methodology is therefore not limited by the clock speed but by the spread condition. In the main text we only explain the relevance of the theorem, while the proof is given in the supplement. We changed the formatting, to avoid any indication of an injected subsection and expanded the discussion of the bounds of the methodology for investigating the high-frequency limits.

18) Is there a way to model the memory effect observed here through non-Markovian rate equations or other tools for instance?

Yes, in principle the memory effect could be modelled by non-Markovian rate equations or by tracking some environmental degree of freedom. We have not attempted this in the paper, because the focus was on the identification of the effect. To further explore its origin and suggest a model calls for a dedicated study as we note in the manuscript: “Both times are significantly longer And raise the need for a dedicated investigation”. The class of behaviors captured by the spread condition can always be described by a Markovian model with time-dependent rates. When this time- (i.e., step-number-) dependence arises not from external modulation of the driving stimulus but from internal memory (as in the experiment reported), such modelling of course, is not revealing the mechanism but merely parametrizes the observed phenomenology of the memory effect.

Given a plausible hypothesis on the physical mechanism of this memory, one can add corresponding additional variables. It is conceivable that in an appropriately extended configuration space, even Markovian dynamics with time-independent transition matrix can generate the observed statistics. If in such a scenario the memory (environmental) degrees of freedom can be traced out, the resulting model will describe non-Markovian stochastic dynamics.

We have refrained from proposing alternative models in the manuscript because our focus is on the ability to detect & quantify the memory effect, while its relation to device physics requires additional investigation to justify model extensions.

19) Note I: As the authors provide the full derivation of the multinomial distribution specific to their experiment, I would suggest to provide all details. To check their derivations, I had to go back to the definition of the multinomial distribution, and find what are the variables. I would suggest to start from the PMF

$$f(k-1, k_0, k_1; t, P_-, P_0, P_+) = \sum_K t! / (k_1! k_0! k-1!) P_-^{k-1} P_0^{k_0} P_+^{k+1}$$

that everyone understands. And then say that you define $s = k-1$.

How do you derive the upper bound for the sum over s , $(t-x)/2$?

You could add the definition of the Pochhammer’s symbol, and provide the intermediate steps. This would also make the identification with the hypergeometric function easier (and you could also provide the steps).

We appreciate the contributions of the referee to make our proof clearer and more straightforward. The proof has been re-formulated as suggested.

You remark that other models have been previously used for this type of circuits. Could you discuss then the advantage of your model compared to those? Did you try to do the same type of analysis and significance test with those models?

This may be a misunderstanding, since we do not explicitly refer to “other models used for this type of circuits”. However, this may have stemmed from our comment in Supplementary Note I about related mathematical works on random walks (“We note that discrete distributions similar to X have been considered before.”). This prior work is relevant only once the abstract representation of the circuit in terms of the random walk (which is the key new concept our paper) has been introduced.

Typos and minor things:

- Page 1: an fixed number of electrons -> a fixed number of electrons

- References to Eqs. do not always appear correctly, see eg. Note I before claim 1 (??). Check throughout the manuscript.

We have fixed the typo and missing references.

Reviewer #3:

3.1) Given the motivation discussed in the introduction (single electron circuit validation) for the study it seems a shame that the proof of this assertion (Tables S1 and S2) is at the end of a very long and dense supplemental information section. I suggest the authors incorporate this information into the main text or at least feature it more prominently in the supplemental.

We have reorganized the Supplemental Notes to better reflect the order of importance and occurrence in the main text. In the now revised and extended description of the experiment in the main text we have added an explicit reference: "Experimental evidence for strong discord between component-wise and circuit-level characterization is given in Supplementary Note II B".

3.2) The manuscript lacks experimental details necessary to understand the experiment. These include, but are not limited to, the detector bandwidth, measurement time, any thresholds used for distinguishing between charge states of the island, how each individual device was tuned, whether the devices were retuned together when combined for error measurements. Why was no magnetic field applied? The above experimental concerns seem important since whether or not the 'x are fully resolved in counting' makes the difference between stationary P+ and P- and non-stationary values. In addition, there was no mention of a numerical value for N except parenthetically in the methods section (this was my first question when examining figures 1 and 2).

The description of the experiment was intentionally concise to emphasize the abstraction of device details. We do agree however, that the arguments for the reliability of the detector are important.

We have changed the manuscript to clarify the experimental details requested by the reviewer:

- New graph illustrating the detector performance has been added as Figure 1b
- Description of the experiment has been expanded to include the explanation of the detector and device operation, experimental conditions and data acquisition timeline.

3.3) The authors should also discuss at more length the viability/accuracy/applicability of the modeling especially at higher frequencies. There is already an effective frequency limit of 150 MHz (300 MHz signal with 3.3 ns delay). Metrologists are certainly interested in higher frequencies than this. Beyond the spread condition, what are the limitations of this model?

As the model probes an abstracted error syndrome of the circuit, the limitation lies in the maximal error rate, which can still be correctly accounted for by the accumulating probe, and, similarly, in the dynamic range of the detector. While approaching this device-specific limit is not the intent of this paper, the methodology shown here to probe and identify the relevant timescales should certainly help to improve circuit performance for metrological applications. To investigate the specific physical origin and suggest development strategies, however, would require a dedicated experiment.

We expanded the discussion of the section on memory time to reflect this aspect.

3.54) There are several missing references in the supplemental and the grammar is sometimes incorrect leading to a lack of clarity.

We have corrected the missing references and revised the supplemental material.

3.5) The manuscript reads like a mathematical exercise without much physical insight. I do not suggest the authors reduce the formal mathematics but rather expand on issues like the speculation

of two-level fluctuators and how these cause “slow” drift in P_{\pm} . What can we say are the characteristics of this drift? The only connection to the physical devices offered is to say that the drift may be due to TLFs. While “fast” and “slow” drift modeling are performed, it seems to me that “fast” and “slow” are relative to the timescale for t steps. Is this correct? If so it seems that the meaning of “fast” and “slow” varies over the course of these experiments. How do we understand the modeling and source of drift in P_{+} and P_{-} in this context?

We have followed the reviewer’s recommendation and expanded the investigation of two-level fluctuators (TLFs) in relation to our statistical tests. We have performed simulations of parametric variability that would be induced by a single or multiple TLFs acting at different timescales and applied our statistical methodology to these artificial data.

The results are presented in the main text (see a new paragraph right before turning to timing and memory effects on device B) and detailed in a new supplementary note (Note V).

Concerning the specific questions of the referee:

The timescale of the test: The physical timeline of data accumulation has not been sufficiently clearly communicated in the original version of the manuscript, which may have led to confusion. The time scale of t steps is typically much shorter than the detection time, hence the timescale of the “fast fluctuator” test is fixed, and the time-scale of “slow drift” depends on the number of samples taken for a fixed burst length. That number, however, does not vary much both for the experimental and the simulated data analyzed. All this is now clarified in the updated version (Figure S1, Supplementary Note V and appropriate references to these in the main text).

Implication for interpreting the test results: Although the timescales of “fast” and “slow” are fixed by the particular timescales for data accumulation in the experiment, we find in simulations that the two tests can discriminate correctly which part of the noise spectrum is dominant. We have tested this in simulations of $1/f$ noise (multiple TLFs) and with a single TLF. In particular, different outcomes of the “fast fluctuator” (reject) and “slow drift” (not reject) tests are observed for a single TLF with a fixed switching rate if the said rate falls between the “fast” and the “slow” timescales (differing by 5 orders of magnitude), thus justifying the names assigned to the models

Implications for the source of the drift: We have found that an ensemble of TLFs with $1/f$ spectrum can produce a statistical picture very similar to the one observed in the experiment. More firm conclusions about the physics of the particular device would require new measurements with deliberately varied physical timescales corresponding to a chunk of data counts that the “slow drift” model treats as generated with fixed single-step probabilities. Our goal in this manuscript has been to provide device-agnostic discovery methodology that enables, in particular, such multiscale approach [see, e.g., Phys. Rev. Appl. 3, 044009 (2015)] to noise spectroscopy.

REVIEWERS' COMMENTS

Reviewer #1 (Remarks to the Author):

The authors revised the manuscript substantially according to the points I raised. The main modifications are summarized as follows.

1) The authors inserted a new paragraph to explain the importance of the proposed error analysis in the presence of long-term drifts and noise, which could access to time correlated errors and memory effects in single-electron pumps beyond the conventional independent error model. Related to the universal gate quantum computation, some papers are cited here as examples of the discussion on the effect of noise correlations on randomized benchmarking.

While I am not still sure if the present study would be readily impactful both to quantum metrology, future quantum circuits, and quantum computers, I can agree that the use of the rigorous statistical testing for this propose is new and would be of potential significance to predict the ultimate accuracy and the scalability of the circuits in the end.

2) To validate their Dirichlet distribution model for more realistic noise environments, the authors conducted new simulations for the system with a single or ensemble two-level fluctuates. The discussion was inserted as a new paragraph to explain that they model can be used for such system and it is beneficial for accurate analysis of the error with long-time correlations.

3) They added Fig. 1 b and discussed that the detection error was low enough to justify their analysis. Considering the substantial change they made with new simulation results as well as the high-level experiments and mathematics they did as noted in my first review, I can recommend this manuscript for Nature Communication.

Note: A part of my question 4) is due to my misunderstanding of the inverted time axis in Fig. 3.

Reviewer #2 (Remarks to the Author):

The authors have answered all my points in a satisfactory way, and I find their answers to all referees complete. Changes have been made accordingly to the manuscript, which I think now meets the standards of Nature Comms. I therefore recommend its publication, without any additional change.

Reviewer #3 (Remarks to the Author):

Second Review of "A random-walk benchmark for single-electron circuits"

I believe the authors have sufficiently addressed my concerns and I can now recommend publication of the manuscript. My more specific responses are below.

3.1) Given the motivation discussed in the introduction (single electron circuit validation) for the study it seems a shame that the proof of this assertion (Tables S1 and S2) is at the end of a very long and dense supplemental information section. I suggest the authors incorporate this information into the main text or at least feature it more prominently in the supplemental.

We have reorganized the Supplemental Notes to better reflect the order of importance and occurrence in the main text. In the now revised and extended description of the experiment in the main text we have added an explicit reference: "Experimental evidence for strong discord between component-wise and circuit-level characterization is given in Supplementary Note II B".

I still believe this should be more prominent but I accept the limitations that the authors discuss regarding the presentation of the material.

3.2) The manuscript lacks experimental details necessary to understand the experiment. These include, but are not limited to, the detector bandwidth, measurement time, any thresholds used for distinguishing between charge states of the island, how each individual device was tuned, whether the devices were retuned together when combined for error measurements. Why was no magnetic field applied? The above experimental concerns seem important since whether or not the 'x are fully resolved in counting' makes the difference between stationary P+ and P- and non-stationary values. In addition, there was no mention of a numerical value for N except parenthetically in the methods section (this was my first question when examining figures 1 and 2).

The description of the experiment was intentionally concise to emphasize the abstraction of device details. We do agree however, that the arguments for the reliability of the detector are important. We have changed the manuscript to clarify the experimental details requested by the reviewer:

- New graph illustrating the detector performance has been added as Figure 1b
- Description of the experiment has been expanded to include the explanation of the detector and device operation, experimental conditions and data acquisition timeline.

I believe the authors have made sufficient changes to the manuscript and included a satisfactory level of detail concerning how the experiment was conducted. I do think the authors are too focused on ensuring a high level of abstraction but the changes made improve the level of clarity.

3.3) The authors should also discuss at more length the viability/accuracy/applicability of the modeling especially at higher frequencies. There is already an effective frequency limit of 150 MHz (300 MHz signal with 3.3 ns delay). Metrologists are certainly interested in higher frequencies than this. Beyond the spread condition, what are the limitations of this model?

As the model probes an abstracted error syndrome of the circuit, the limitation lies in the maximal error rate, which can still be correctly accounted for by the accumulating probe, and, similarly, in the dynamic range of the detector. While approaching this device-specific limit is not the intent of this paper, the methodology shown here to probe and identify the relevant timescales should certainly help to improve circuit performance for metrological applications. To investigate the specific physical origin and suggest development strategies, however, would require a dedicated experiment.

We expanded the discussion of the section on memory time to reflect this aspect.

I'm inclined to agree with the authors. I had hoped that there would be some comments the authors could make about how difficult it would be to achieve then necessary uncertainty in the model parameters when the pump is accurate and operated at high frequency. The authors point out that, in principle, the presented error syndrome can still correctly account for errors in the regime metrologists are interested in. I now agree that further questions are beyond the scope of this paper.

3.S4) There are several missing references in the supplemental and the grammar is sometimes incorrect leading to a lack of clarity.

We have corrected the missing references and revised the supplemental material.

Agreed

3.5) The manuscript reads like a mathematical exercise without much physical insight. I do not suggest the authors reduce the formal mathematics but rather expand on issues like the speculation of two-level fluctuators and how these cause "slow" drift in P+/- . What can we say are the characteristics of this drift? The only connection to the physical devices offered is to say that the drift may be due to TLSs. While "fast" and "slow" drift modeling are performed, it seems to me that "fast" and "slow" are relative to the timescale for t steps. Is this correct? If so it seems that the meaning of

"fast" and "slow" varies over the course of these experiments. How do we understand the modeling and source of drift in P+ and P- in this context?

We have followed the reviewer's recommendation and expanded the investigation of two-level fluctuators (TLFs) in relation to our statistical tests. We have performed simulations of parametric variability that would be induced by a single or multiple TLFs acting at different timescales and applied our statistical methodology to these artificial data.

The results are presented in the main text (see a new paragraph right before turning to timing and memory effects on device B) and detailed in a new supplementary note (Note V).

I appreciate the authors work in response to this. Unfortunately, I do not see much physical insight gained by the extra work performed. At most, it seems to me that the data can be found to be more consistent with a single fluctuator ($1/f^2$) or an ensemble of them ($1/f$). I don't suggest changing anything further. I have convinced myself that it is beyond the power of this modeling to tell us anything more than what the authors have already told us.

Concerning the specific questions of the referee:

The timescale of the test: The physical timeline of data accumulation has not been sufficiently clearly communicated in the original version of the manuscript, which may have led to confusion. The time scale of t steps is typically much shorter than the detection time, hence the timescale of the "fast fluctuator" test is fixed, and the time-scale of "slow drift" depends on the number of samples taken for a fixed burst length. That number, however, does not vary much both for the experimental and the simulated data analyzed. All this is now clarified in the updated version (Figure S1, Supplementary Note V and appropriate references to these in the main text).

I appreciate the increase in clarity brought by the comments. I think I am still correct that the timescale of the test varies – it just does not vary much so my comment above is not very important.

Implication for interpreting the test results: Although the timescales of "fast" and "slow" are fixed by the particular timescales for data accumulation in the experiment, we find in simulations that the two tests can discriminate correctly which part of the noise spectrum is dominant. We have tested this in simulations of $1/f$ noise (multiple TLFs) and with a single TLF. In particular, different outcomes of the "fast fluctuator" (reject) and "slow drift" (not reject) tests are observed for a single TLF with a fixed switching rate if the said rate falls between the "fast" and the "slow" timescales (differing by 5 orders of magnitude), thus justifying the names assigned to the models

Agreed. See above

Implications for the source of the drift: We have found that an ensemble of TLFs with $1/f$ spectrum can produce a statistical picture very similar to the one observed in the experiment. More firm conclusions about the physics of the particular device would require new measurements with deliberately varied physical timescales corresponding to a chunk of data counts that the "slow drift" model treats as generated with fixed single-step probabilities. Our goal in this manuscript has been to provide device-agnostic discovery methodology that enables, in particular, such multiscale approach [see, e.g., Phys. Rev. Appl. 3, 044009 (2015)] to noise spectroscopy.

Agreed. See above